# On the Optimal Indoor Air Conditions for SARS-CoV-2 Inactivation. An Enthalpy-Based Approach

**DOI:** 10.3390/ijerph17176083

**Published:** 2020-08-21

**Authors:** Angelo Spena, Leonardo Palombi, Massimo Corcione, Mariachiara Carestia, Vincenzo Andrea Spena

**Affiliations:** 1Department of Enterprise Engineering, Tor Vergata University of Rome, 00133 Rome, Italy; spena@uniroma2.it; 2Department of Biomedicine and Prevention, Tor Vergata University of Rome, 00133 Rome, Italy; palombi@uniroma2.it; 3Department of Astronautical, Electrical and Energy Engineering, Sapienza University of Rome, 00184 Rome, Italy; massimo.corcione@uniroma1.it; 4Department of Industrial Engineering, Tor Vergata University of Rome, 00133 Rome, Italy; mariachiara.carestia@uniroma2.it

**Keywords:** SARS-CoV-2, CoViD-19 pandemic, indoor air quality, specific enthalpy of moist air, HVAC systems setting

## Abstract

In the CoViD-19 pandemic, the precautionary approach suggests that all possible measures should be established and implemented to avoid contagion, including through aerosols. For indoor spaces, the virulence of SARS-CoV-2 could be mitigated not only via air changes, but also by heating, ventilation, and air conditioning (HVAC) systems maintaining thermodynamic conditions possibly adverse to the virus. However, data available in literature on virus survival were never treated aiming to this. In fact, based on comparisons in terms of specific enthalpy, a domain of indoor comfort conditions between 50 and 60 kJ/kg is found to comply with this objective, and an easy-to-use relationship for setting viable pairs of humidity and temperature using a proper HVAC plant is proposed. If confirmed via further investigations on this research path, these findings could open interesting scenarios on the use of indoor spaces during the pandemic.

## 1. Introduction

The outbreak of the SARS-CoV-2 pandemic, having a huge impact on occupational and public health and safety, posed much concern on the spread of the virus in confined environments, especially in inhabited spaces. Although specific and effective therapies and vaccines are under development, other factors that can help to prevent the spread of the disease have to be considered, and proper mitigation measures need to be implemented. It has been recently inferred that good mechanical room ventilation using outdoor air would have an effect similar to a 50–60% vaccination coverage in a poor ventilation scenario [1]. According to one of the first experimental studies available in literature on the SARS-CoV-2 virus, the stability of the virus in air and on surfaces is very similar to that of SARS-CoV-1 [2]. Therefore, SARS-CoV-2 viability in aerosol, depending on places, circumstances, and human factors, cannot be neglected [3,4,5,6]. Such a statement, unlike what has been stated in former studies [7], is true especially for indoor spaces, which leads to an assumption that the infectivity of the virus may be possibly reduced if indoor hygrothermal conditions hostile to the virion could be properly established using a suitable HVAC (heating, ventilation, and air conditioning) system [8]. Nonetheless, the pursuit of this objective must consider a number of other aspects: (i) the vulnerability of human mucous membranes tends to increase in decreasing relative humidity (RH) below 30% [9]; (ii) the presence of humans inside indoor spaces for a long time (many hours) requires adequate, or at least bearable, comfort conditions [10]; (iii) to avoid the potential for mould and moisture threat, RH cannot exceed specific values (at most 80%) in the coldest and least ventilated corners [11]; and (iv) to offset the risk of electrical static charge on electronic devices in office or healthcare environments, RH cannot be extremely low [12]. All of these statements highlight the need to identify a domain of RH and temperature values that could result in a decrease in the virus strength, being at the same time suitable for long-term human presence, building-structure preservation, and critical-devices operation [13]. Indeed, although dedicated HVAC systems can control indoor humidity and temperature, which can have direct effects on the transmission of infectious agents, a conclusive and unique recommendation is not yet available in literature [14,15,16,17,18], because existing studies have focussed on different objectives and have been conducted mostly using a solely virological approach [19,20,21,22]. The lacking facets of the research concern transmissibility not only via surfaces and fomites and aerosol droplets, but also, to a minor extent, via airborne particles, even though the latter could probably be more successfully inhibited than the former [6,23,24]. This study represents the first of an interdisciplinary approach to define the best environmental conditions that should be ensured by HVAC systems to oppose the spread of CoViD-19 in any indoor inhabited space. Despite the scarcity of available in-field studies to date, this study also enables us to infer some general considerations regarding this approach to viral spread prevention.

## 2. Material and Methods

### 2.1. Background

Viruses are among the smallest of the common primary biological aerosol particle (PBAP) classes, with physical diameters generally ranging from 20 to 250–400 nm [25]. They are commonly not airborne as individuals and are more likely to be attached to other suspended particles [26,27]. In the past two decades, new zoonotic coronaviruses have emerged, causing outbreaks in humans [28], i.e., SARS-CoV (2002, Betacoronavirus, subgenus Sarbecovirus) and MERS-CoV (2012, Betacoronavirus, subgenus Merbecovirus). In late 2019, a novel coronavirus related to a cluster of pneumonia cases in Wuhan, China (SARS-CoV-2), closely related to SARS-CoV and genetically clustered within the Betacoronavirus subgenus Sarbecovirus [29,30,31], was identified. SARS-CoV and MERS-CoV share relatively similar stability characteristics [32].

In the literature, the incidence of illness and infectivity of a virus transmitted via the airborne route in an indoor environment is acknowledged as a result of many factors [20,21]. These factors include humidity, particle size, temperature, population density, number of susceptible hosts, length of exposure, number of infected people producing contaminated aerosols, ventilation rate, infectious particle settling rate, whether the virus is enveloped or not, the presence of other aerosols or volatile organic compounds (VOC), exposure to UV light or to chemicals that inactivate or kill the virus, among others. Because nanometric particles carry the virus by forming wet droplet nuclei [33], virus aerosolisation in any confined space, mainly if the spread is from asymptomatic infected persons, may be dangerous via indirect transmission [3], including due to the resuspension of floor dust [34,35,36]. This nature of viruses reinforces the importance of avoiding crowded gatherings and implementing early identification and diagnosis of asymptomatic carriers for early quarantine or treatment [17].

Moreover, the concentration and infectivity of smaller airborne particles decrease more slowly, and thus their longer retention time in the air emphasises the concern that they pose. Both are applicable to SARS-CoV-2, which has a size in the order of 20 nm. In 2010, Hanley and Borup stressed [37] that, in addition to intervention strategies including the use of masks and gloves, climate control of indoor locations should be considered by public health planners in developing recommendations to interrupt the spread of influenza. Accordingly, it should be remembered that the initial 187 cases of SARS-CoV-1 in the housing complex of Amoy Gardens in Hong Kong (2003) were related, with a reasonable degree of certainty, to bioaerosolisation [23]. Based on the consideration stated, a reasonable precautionary approach to the CoViD-19 pandemic is to be attempted, by all means, to prevent infections via aerosol [24].

### 2.2. Literature Synopsis on Contributing Factors

#### 2.2.1. Virus Survival and Transmission

Relative Humidity RH—three mechanisms, even acting in a somewhat concerted manner, could explain the observed influence of RH on transmission [6]:(i)RH may act at the level of the environment. Since a higher humidity implies a slower evaporation from large droplets, the faster evaporation occurring at low RHs would more rapidly change these into droplet nuclei (<5 micron), quickly enough before they fall. Thus, people inhale fewer droplets at a higher RH.(ii)RH may act at the level of the host. The activity of nasal mucosa strongly depends on the humidity and temperature of the inhaled air, on the exposure time, and on the health of the individual [38]. Lower or higher RH, compared to medium RH values, will alter the mucous viscosity and mucociliary activity. In general, extremely low humidities are reported as enabling the viruses’ settlement in human hosts. Moreover, the dryness of the respiratory epithelium, which plays an important role via the evaporation of water from its surface (desiccation), may increase bacterial adherence and allows for greater penetration of foreign species, such as particles [39]. The upper airways need to achieve moisture neutrality and maximum mucociliary transport as fast as possible. However, the corresponding saccharin mucociliary clearance time in the upper airways is a function of RH [9]. This leads to a recommendation of RH > 30% to avoid “dry eyes”, and RH > 10% to avoid nasal dryness.(iii)RH may act at the level of the virus particle, affecting its virulence. Virus stability in air may directly affect virus transmission, because virus particles need to remain viable long enough after being expelled from the host to be taken up by a novel host (indirect transmission) [3].

Absolute Humidity (AH)—RH does not provide a fixed measure of water vapour content in air. Virus transmission responds to the amount of water vapour in the surrounding dry air, i.e., AH, and not how close that air is to saturation, i.e., RH [20]. Many studies have shown that the survival and transmission potentials of influenza viruses in wintertime are inversely associated with AH rather than with RH [22,40,41,42,43], and that low temperature and low AH, as opposed to higher temperature and humidity, prevents the disruption of the influenza virus [44,45]. Indeed, lipid-containing viruses are generally more stable in aerosols and are more labile in moist air than in dry air [46,47], although the presence of proteins can alter this relationship [48].

Temperature—the role of temperature appears to be more complex, but relatively less heavy than those other factors. It was suggested that under the conditions of high humidity, the fluidity of lipid-containing envelope is stabilised at low temperature, thereby protecting the virion [49]. Cold temperatures and low RH conditions favour the survival and transmission of certain influenza virus [50,51], and are associated with an increased occurrence of respiratory tract infections.

Concentration of salts (pH)—as salts become more concentrated as water is lost through evaporation [4], it has been hypothesised that RH can affect the viability of enveloped viruses by altering the pH in the aerosols [14], which, in turn, induces conformational changes to the viral glycoproteins and damages to viral infectivity [3]. Viability decreases in saline solutions, but does not change significantly in solutions supplemented by proteins, and increases dramatically in mucous [6,15].

#### 2.2.2. Indoor Environment Vulnerabilities

Domestic comfort and work performance—indoor air quality (IAQ) is often perceived as more acceptable at low RH and low temperature [10,52], depending on alterations not only of the VOC emission profile, but also of dynamics, composition, deposition and resuspension of inhaled particles. However, this contrasts the outcome of many studies that show that elevated RH may reduce complaint rates, and supports eye tear-film stability and physiology (as already mentioned) and osmolarity of the upper airways, thus enhancing work performance in offices. Furthermore, laboratory experiments reveal a greater resuspension of large particles and lower resuspension of smaller particles at low RH [18,36]. As a reference, the ASHRAE (American Society of Heating, Refrigerating and Air-conditioning Engineers) recommended comfort zone for domestic and office-like environments [13] is highlighted in Figure 1, in which the psychrometric chart of moist air at atmospheric pressure is displayed.

Building salubrity and equipment safety—when building materials or furnishings are damp for a consistent time period, mould and bacteria often colonise the materials, and can produce microscopic airborne particles in the indoor air [53,54], some containing allergens or chemicals with the potential to induce inflammation in the respiratory system [6]. Typical isopleths of mould germination and growth rates reported in Figure 2 show that the wetter is a surface, the quicker mould spores will germinate and grow [55,56]. Typical room temperatures of 21–22 °C are ideal for mould growth. Whilst WHO recommends an indoor RH not higher than 45% to inhibit the growth of mites [11], according to Figure 2 a reasonable limit that balance conflicting requirements appears to be 80%. Furthermore, electric or electronic devices consistently require higher humidities to reduce electrical static charge, which may otherwise occur in medical test and therapy equipment, as well in computer areas, control halls and data centres.

Hospital epidemiology and HVAC setting—nosocomial transmission has been widely described as an important driver in the epidemiology of SARS and MERS [57]. A notable feature of the SARS disease was its predilection for transmission in the health care setting and to close family and social contacts. During the outbreak of SARS-CoV-1 in Guangzhou, clinicians kept the windows of patient rooms open and well ventilated and this was thought to have reduced virus survival, thereby reducing nosocomial transmission [19]. Healthcare workers treating influenza patients are particularly prone to infection because they can be exposed to aerosolised viruses from multiple patients in closed clinical environments during the flu seasons [58]. This led to the ASHRAE recommendation of identifying cohorting possibilities for pandemic situations for whole areas of a hospital to be placed under isolation and negative pressure [6]. In fact, negative pressure ventilation and high air exchange rate were effective in minimising airborne SARS-CoV-2 inside the intensive care unit (ICU), coronary care unit (CCU) and ward rooms of Renmin Hospital, where sub-micron aerosols containing the virus were hypothesised to come from the resuspension of virus-laden aerosols from staff apparel because of its higher mobility [59].

### 2.3. Challenging Trade-Off

In outdoor open spaces, social distancing can be considered a good prescription for preventing viral infections. However, for confined indoor environments, whose dimensions limit both the distance between the people and the dilution of the virus concentration, social distancing seems less applicable and effective [60], and thus a different approach is required. The progress in implementing clean, even aseptic, confined spaces shows the level of capability that environmental engineering has reached throughout the past decades; indeed, today it is possible to set, control and maintain a number of environmental parameters in confined and crowded spaces in industrial, residential and transportation applications by means of proper HVAC equipment [61,62]. There are two possible ways of achieving this result. One is quantitative, i.e., at the source level, based on the dilution of the infectious agent load via air changes [63,64,65], their capture through devices such as ultra-low particulate air (ULPA) filters or electrostatic precipitators (ESPs) [12], or the use of UV radiation [6]. The other is qualitative, i.e., at the ambient level, reachable through the surrounding of the infectious agents with adverse thermodynamic conditions. The present paper expressly deals with only the second action, though without any prejudice to the concurring use of some or all the other quantitative actions. As discussed earlier, the infectious mechanisms are far from being fully understood, thus generalisations are not applicable, and solutions should be dealt with on a nearly virus-by-virus basis [8,48]. This could be due not only to the inadequacy or scarcity of the available experimental studies, or to the overall complexity of the phenomena as reported in literature [14,20,66], but also probably to intrinsic differences in the families of influenza and CoV viruses, and mainly to the lack of a holistic approach by the researchers. In fact, cold and dry conditioned air in office-like environments appears to favour survival and transmission of certain airborne viruses, but not of others [48,67,68]. Moreover, when selecting indoor temperature and humidity levels, the aforementioned concurrent aspects need to be balanced, in relation to both human comfort and to building salubrity and equipment operation [66,69,70], provided that a proper HVAC system, rather than mere split air-conditioners, or similar, with no air renewals [62], are available.

### 2.4. A Parameter for Correlating Literature Data

The experimental data available on aerosolised coronavirus survival are very few and widely scattered throughout the temperature–humidity phase space, which indicates that their analyses cannot lead to any definitive result, yet it may delineate a path for future investigation on SARS-CoV-2. Only four studies containing useful information pertaining to this topic have been readily found in open literature, which allowed us to evaluate the viability of CoV virus after aerosolisation and put it in direct relationship with the environmental conditions. The available data report survival rate over time in terms of either Plaque Forming Units (PFU) or Median Tissue Culture Infectious Dose (TCID_50_), whose values can be put in relation to each other, provided that given assumptions on the cellular line used and the titration protocols are verified [71]. However, for the purpose of the present study, in order to compare the survival data from different sources expressed with different units, we evaluated the survival rate reduction over time (1 h) for different RH and temperature parameters, by introducing a parameter called Viral Load Survival, denoted as VLS_1h_, with the general meaning of “percentage of detected levels of virus survival after one hour from spread” and calculated as the ratio between the viral load after one hour and that at the moment of its diffusion, regardless of the method used by the different research teams to determine the infectious titre of the virus, i.e., PFU or TCID_50_.

First of all, the results reported by Prussin et al. [72], who investigated the survival of the enveloped bacteriophage Phi6, have been considered as a reference. In fact, Phi6 can be treated as a reliable surrogate of CoV-1 coronaviruses in the estimation of their survival, see Adcock et al. [73] and Turgeon et al. [74]. Phi6 survival after 1 h has been investigated both in the form of droplets and aerosol, with respect to RH at a temperature of 22 °C, resulting in a correlation coefficient of 0.98 and no significant difference between the two distributions (*p*-value = 0.095). The collected data, expressed as relative infectious ratio, i.e., the ratio of the concentration of PFU derived from exposed samples to the concentration of PFU in the control sample, pointed out the existence of a remarkable minimum for VLS_1h_ of nearly 1% at 75% RH. Notice that a significant decrease in infectivity at mid-range RHs was reported also in previous investigations looking at coronaviruses’ survival on surfaces, such as, for example, the study carried out by Casanova et al. [75].

Subsequently, the data available from the papers published by van Doremalen et al. [76], and Pyankov et al. [77], have been considered. Both research groups analysed the aerosol stability of human betacoronavirus EMC (isolate HCoV-EMC/2012) assumed as representative of MERS-CoV-1, reporting the time-distributions of TCID_50_ as a function of RH and temperature. Van Doremalen et al. [76] reported also the aerosol stability of H_1_N_1_ (A/Mexico/4108/2009).

From the first study, values of VLS_1h_ of 93% at 20 °C and 40% RH, and of 11% at 20 °C and 70% RH, can be extracted. From the second study, values of VLS_1h_ of approximately 63% at 25 °C and 79% RH, and of 4.7% at 38 °C and 24% RH, can be determined. The latter value can be determined to be susceptible to an increase from 4.7% to about 25%, owing to a potential underestimation, which can be easily deduced from an attentive inspection of the time-trend displayed for the airborne virus concentration, compared with the expected exponential decay extrapolated using all the virus concentrations measured in the first 30 min. Additionally, a couple of datasets extracted from a recent investigation conducted by van Doremalen et al. [2] at 22 ± 1 °C and 65% RH using both SARS-CoV-1 and SARS-CoV-2, have also been counted. According to the reported time distributions of TCID_50_, virus survivals of nearly 29% and 30% can be estimated for SARS-CoV-1 and SARS-CoV-2, respectively.

Notice that only Pyankov et al. [77] declared tolerances in thermoigrometric measurements, namely ±0.5 °C in temperature and ±2% in RH. According to Noti [58], data were generally considered significant if *p*-value < 0.05.

## 3. Results

To correlate all the collected data (see Table 1), for each thermodynamic state of equilibrium identified by its temperature and relative humidity, the specific enthalpy of moist air h has been calculated as follows [78]:h = c_a_t + AH (c_v_t + r)(1)
with
AH = 0.623 [(RH%/100) p_s_(t)]/[p − (RH%/100) p_s_(t)](2)
where c_a_ and c_v_ are the specific heats at constant pressure of dry air and water vapour, which, around ambient temperature, can be assumed as equal to 1.006 kJ/kg °C and 1.86 kJ/kg °C, respectively; t is the temperature in degrees centigrade; AH is the absolute humidity of moist air, in kg_v_/kg_dry-air_, also called humidity ratio and defined as the ratio of the mass of water vapour to the mass of dry air in the moist air sample; r is the latent heat of vapourisation of water at its triple point, equal to 2501 kJ/kg; p_s_(t) is the saturated vapour pressure of water at temperature t in Pascal, and p is the total pressure of moist air, typically the atmospheric pressure, in Pascal.

The saturated vapour pressure of water in Pascal can be calculated from the empirical formula derived by Hyland and Wexler for the temperature range of 0 to 200 °C [78,79]:lnp_s_(T) = C_1_/T + C_2_ + C_3_T + C_4_T^2^ + C_5_T^3^ + C_6_lnT(3)
in which C_1_ = −5.8002206 × 10^3^, C_2_ = 1.3914493 × 10^0^, C_3_ = −4.8640239 × 10^−2^, C_4_ = 4.1764768 × 10^−5^, C_5_ = −14452093 × 10^−8^, C_6_ = 6.5459673 × 10^0^, whereas T is the absolute temperature in Kelvin degrees, T = t + 273.15.

The distributions of the Viral Load Survival VLS_1h_ of the viruses plotted against the specific enthalpy of moist air are reported in Figure 3, in which the interpolation curve of the reference data of Phi6 bacteriophage is also depicted. It is apparent that the data available for MERS-CoV-1, SARS-CoV-1 and SARS-CoV-2 viruses are substantially well distributed along the curve representative of the behaviour of Phi6 bacteriophage. In this connection, it is worth noticing that, should the survival reported by Pyankov and co-workers for the HCoV-EMC virus at 38 °C and 24% RH be increased from 4.7% to 25%, as discussed earlier, the data for these two sets of conditions would be closer to the interpolation curve. Furthermore, it seems meaningful that for environmental conditions whose specific enthalpy of moist air falls within the range between 50 and 60 kJ/kg-dry air, the survival of all the considered viruses sinks dramatically. This also seems to be in line with recent observations reported regarding the temperature and humidity weather-data recorded along the path going from East to West of the significant spread of CoViD-19 caused by SARS-CoV-2 [80].

## 4. Discussion

### 4.1. Obtained Space of Viable Solutions—Preliminary Recommendations

As a matter of data reliability, errors come from either titration (as TCID_50_) [2,47,48] or infectivity estimation (as PFU) [34], and from temperature and RH recordings, information about the latter being almost lacking in any study cited earlier. However, if we assume an error of ±0.25 °C for the measured temperature, and of ±5.0% RH for the measured RH as realistic for an actual room environment, the consequent uncertainty for the calculated specific enthalpy h of moist air would remain in the order of a few percent, slightly raising from ±4.0% at 20 °C to ±4.7% at 25 °C. This enables us to infer that the proposed interpolation curve portrayed in Figure 3 well complies with the uncertainties intrinsic with the collected data. Notice that no direct correlation has been found to hold between VLS_1h_ and either air temperature t or relative humidity RH% or absolute humidity AH independently of one another, yet a clear relationship seems to exist between VLS_1h_ and the thermodynamic potential specific enthalpy h of moist air, the minimum of which should correspond to optimal pairs of temperature and relative humidity values for coronavirus viral load inactivation, wherein SARS-CoV-2 infectivity actually appears to be nearly suppressed. Even if the recent SARS-CoV-2 could evolve in the future, the literature data show that if the specific enthalpy h of moist air is kept in the range between 50 and 60 kJ/kg-dry air, the virus infectivity appears to be nearly offset. Additionally, as discussed earlier, the relative humidity RH should be maintained neither lower than 30%, for fitness and safety reasons, nor higher than 80%, for mould germination reasons.

The intersection of the region including all the thermodynamic states of moist air such that the specific enthalpy lies in the range 50−60 kJ/kg-dry air and the relative humidity ranges between 30% and 80% with the ASHRAE recommended comfort zone for domestic and office-like environments, coloured in dark grey in the psychrometric chart depicted in Figure 4, identifies the space of viable solutions, i.e., the set of indoor environmental conditions able to satisfy both requirements of SARS-CoV-2 inactivation and hygrothermal comfort of resident people.

Notice that the obtained relationship between the parameter VLS_1h_ and the specific enthalpy of moist air should be strictly applied within the temperature range of investigation, which spans from 20 to 25 °C, as discussed earlier. This, however, as shown in Figure 4, does not represent a true limit of applicability, because—quantitatively—the lower boundary temperature of the ASHRAE recommended comfort zone does practically coincide with 20 °C, while its upper limit does not significantly exceed 25 °C; but also—qualitatively—the observed tendency (a decreasing RH at an increasing temperature) fits well with the usual comfort requirements.

As a preliminary recommendation, we can conclude that the optimum thermodynamic conditions at which moist air should be maintained to reduce the exposure risk in indoor spaces are those confined within the intersection region located in the upper right end of the standard ASHRAE comfort zone. This means that the typical summertime design conditions of 25–26 °C with a 50% RH are located within this region, although very close to its lower boundary, whereas those typical of wintertime, i.e., 20 °C with a 50% RH, lie outside it. Accordingly, it would be advisable that AH of supply air is kept slightly higher than usual, thus implying a larger humidification in winter and a smaller dehumidification in summer, which result in an increase in the heating costs in winter and a decrease in the cooling costs in summer. Moreover, the indoor temperature in winter should be kept some degrees higher than usual, with further additional heating costs.

### 4.2. HVAC Plants Optimal Setting

For any assigned value h of the specific enthalpy of moist air, the relationship existing between relative humidity RH% and temperature t can be obtained by replacing Equation (2) into Equation (1), which, after some algebra, gives r = 2501 kJ/kg, where, as said, c_a_ and c_v_ can be assumed as equal to 1.006 kJ/kg °C and 1.86 kJ/kg °C, respectively.
RH% = 100[p/p_s_(t)]/[1 + 0.623(c_v_t + r)/(h − c_a_t)](4)

This means that, when imposing the optimal value h = 55 kJ/kg-dry air (or any other value of h included in the range between 50 and 60 kJ/kg-dry air), the value of the relative humidity corresponding to an assigned value of the air temperature can be calculated through Equation (4), which leads to identifying the optimal indoor environmental conditions for SARS-CoV-2 inactivation.

As a ready-to-use tool, Figure 5 allows the determination, for an assigned value of the specific enthalpy of moist air h, of the pairs of suitable values of t and RH to be established to reduce the exposure risk inside indoor spaces, directly obtained by way of Equation (4).

## 5. Conclusions

The obtained results confirm that, in general, temperature or humidity cannot be independently correlated with coronaviruses viability, because a one-to-one approach reveals an intrinsically erratic nature. Simultaneously, our results infer a relationship between coronaviruses survival and the thermodynamic potential specific enthalpy of moist air, exhibiting a minimum value for the proposed parameter VLS_1h_ at around 55 kJ/kg-dry air. This appears to unlock the previously acknowledged impasse on research method, thereby opening a broad perspective considering straightforward applications.

We pointed out that a space of viable solutions does exist, representing the set of indoor environmental conditions able to satisfy both requirements of SARS-CoV-2 inactivation and hygrothermal comfort of resident people. This can be easily summarised by the enforcement of a specific enthalpy around 55 kJ/kg-dry air. To further analyse this value, an easy-to-use diagram for determining the optimal pairs of air temperature and relative humidity is proposed. As a matter of fact, the combinations of air temperature and relative humidity optimal for virus inactivation fall not far from those typical for human comfort. Furthermore, conditions preventing extremely low values of RH for sanitary cautions like eyes dryness, and for not too high values of RH for avoidance of mould risks and evapo-transpiration discomfort, appear not to significantly reduce the space of viable solutions.

If our results should be corroborated by further experiments, possibly extended to air temperatures higher than 25 °C and slightly lower than 20 °C, the previously discussed exposed enforcement by means of suited HVAC plant could yield an important in-field support to the prevention and control of CoViD-19 transmission, particularly in the case of recurrence of the disease.

The outlined interdisciplinary methodology can be extended to any health-strategic approach. From an epidemiological point of view, deeper insights could explore and explain the reasons that cause the observed narrow interval of specific enthalpy h of moist air to be critical, whose nature of state property involving overall heat (sensible + latent) seems to suggest some kind of thermodynamical “resonance”, or a melt of crucial factors. More specifically, as regarding the envelope of the virus, a topic to investigate could be specific enthalpy values in the range between 3.9 and 5.1 kJ/g of water vapour, seeking possible cross-linking reactions occurring between the surface proteins of the virus, or certain crystallisation of the virus at the point of efflorescence. Another topic to be addressed could be the mechanism by which salts affect coronaviruses viability at a molecular level, or how the presence of proteins can alter this relationship, searching for singularities in the infectious mechanism of SARS-CoV-2.

This research did not receive any specific grant from funding agencies in the public, commercial, or not-for-profit sectors.

## Figures and Tables

**Figure 1 ijerph-17-06083-f001:**
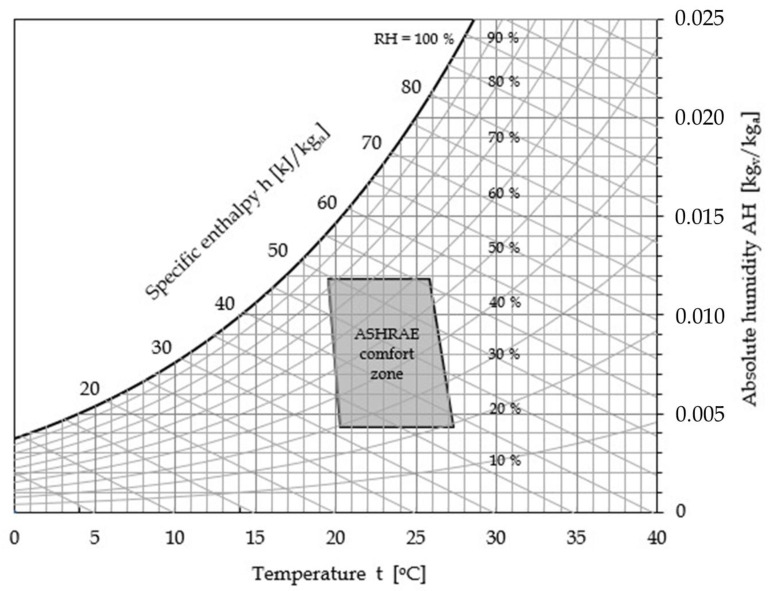
Conventional domestic and office-like comfort zone based on ASHRAE [13].

**Figure 2 ijerph-17-06083-f002:**
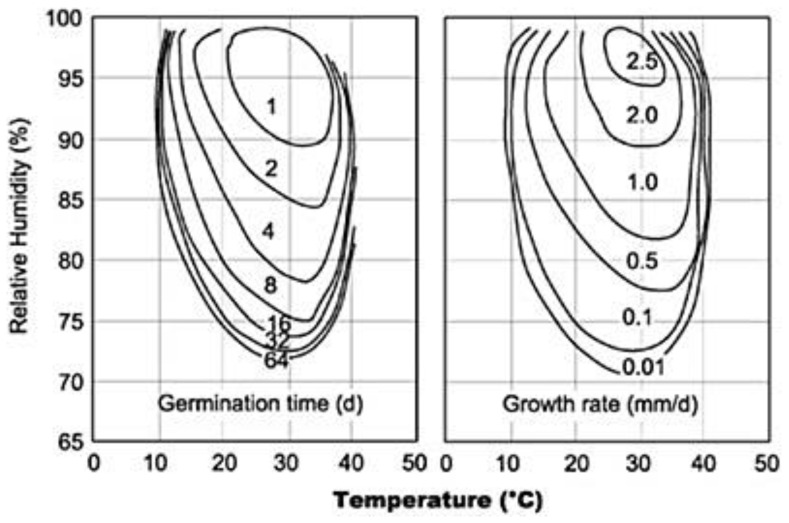
Mould germination and growth rate as a function of t and relative humidity (RH) [56].

**Figure 3 ijerph-17-06083-f003:**
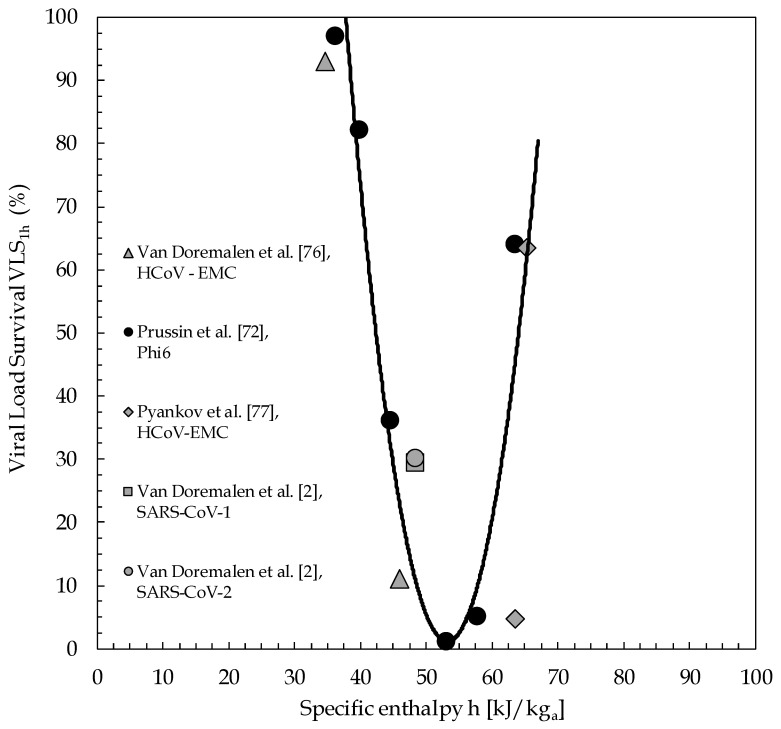
Viral load survival (VLS)_1h_ as a function of specific enthalpy h of moist air (kJ/kg-dry air).

**Figure 4 ijerph-17-06083-f004:**
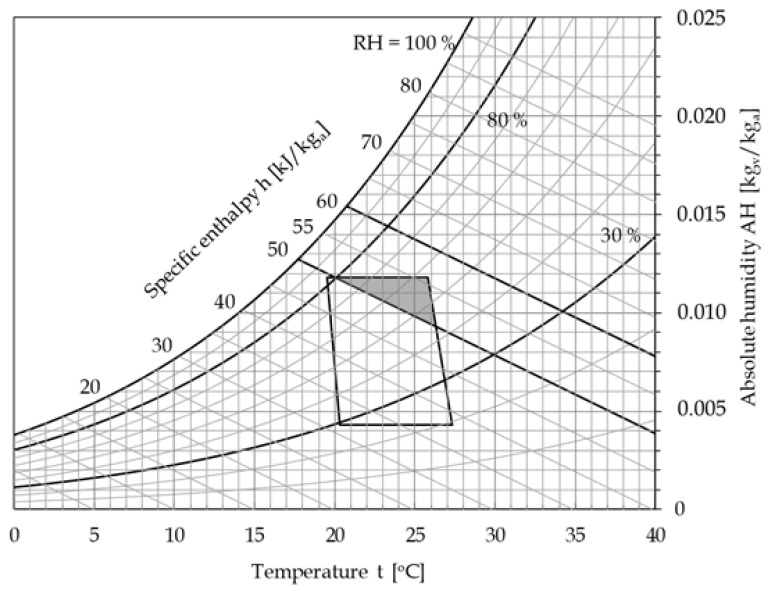
Optimal thermodynamic conditions of moist air to be ensured in indoor spaces to reduce the exposure risk.

**Figure 5 ijerph-17-06083-f005:**
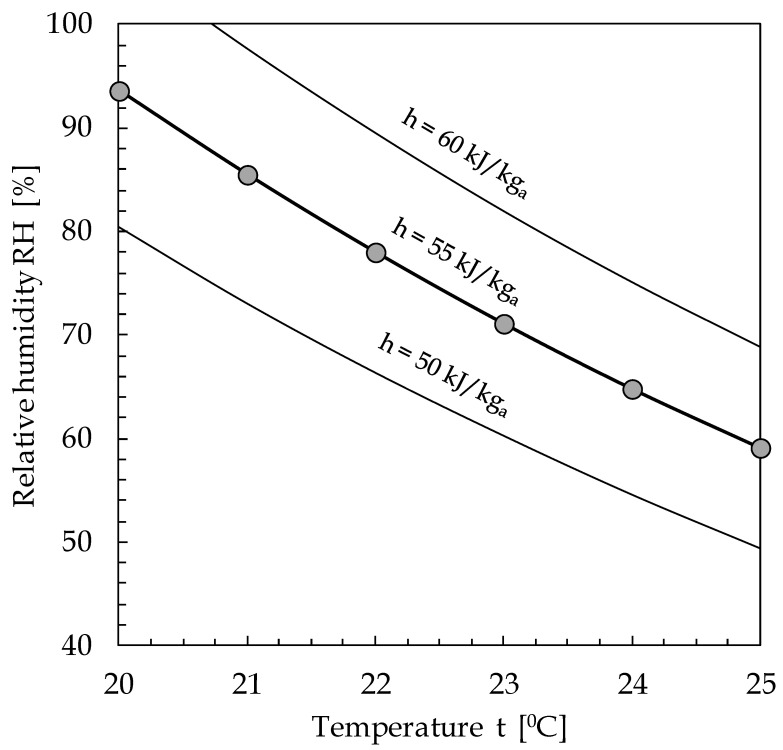
Indoor environmental optimal pairs of t and RH for coronaviruses viral load inactivation.

**Table 1 ijerph-17-06083-t001:** Virus survival obtained from literature experimental data and related information.

	Temperature [°C]	RH (%)	VLS_1h_	Virus Type	AH [kg_vap_/kg_dry-air_]	H [kJ/kg_dry-air_]
Pyankov et al. [77]-2018	25	79	0.634	HCoV-EMC	0.0158	65.32
Pyankov et al. [77]-2018	38	24	0.046	HCoV-EMC	0.0099	63.77
Van Doremalen et al. [2]-2020	22	65	0.300	SARS-CoV-2	0.0107	49.44
Van Doremalen et al. [2]-2020	22	65	0.293	SARS-CoV-1	0.0107	49.44
Van Doremalen et al. [76]-2013	20	40	0.930	HCoV-EMC	0.0058	34.84
Van Doremalen et al. [76]-2013	20	70	0.110	HCoV-EMC	0.0102	46.06
Prussin et al. [72]-2018	22	33	0.970	Phi6	0.0054	35.88
Prussin et al. [72]-2018	22	43	0.820	Phi6	0.0071	40.09
Prussin et al. [72]-2018	22	55	0.360	Phi6	0.0091	45.18
Prussin et al. [72]-2018	22	75	0.010	Phi6	0.0124	53.72
Prussin et al. [72]-2018	22	85	0.050	Phi6	0.0141	58.03
Prussin et al. [72]-2018	22	98	0.640	Phi6	0.0163	63.66

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
