# Peer review of "On the Optimal Indoor Air Conditions for SARS-CoV-2 Inactivation. An Enthalpy-Based Approach"

_ijerph, 2020, doi:10.3390/ijerph17176083_

Round 1
Reviewer 1 Report
Just a couple of very minor comments:
- Could the authors add additional comments on Figure 4? This indicates that the optimum conditions to reduce exposure risk are definitely at the higher end of the standard ASHRAE comfort zone. Are there implications for changes to building HVAC controls & equipment, such as less dehumidification (or possibly more humidification)?
- Some research looking at virus survival on surfaces may further support the ideas in this paper. For example in: Casanova LM, Jeon S, Rutala WA, Weber DJ, Sobsey MD. Effects of air temperature and relative humidity on coronavirus survival on surfaces. Appl Environ Microbiol. 2010;76:2712–2717. May 2010. http://dx.doi.org/10.1128/AEM.02291-09.
- Regarding the English language changes required. While the vast majority of the paper only needs minor spelling and grammar tweaks there are a few places where editing the language would increase the clarity. A little collaboration between the authors and a native English speaker would be helpful.
Author Response
The authors wish to thank the Reviewer for her/his valuable comments and suggestions.
The manuscript has been modified accordingly, including some additional changes useful to improve the clarity of the paper.
The answers to each point raised by the Reviewer are reported below.
- Additional comments on Figure 4, related to the fact that the optimum thermodynamic conditions to reduce the exposure risk in indoor spaces are located in the upper right end of the standard ASHRAE comfort zone recommended for domestic and office-like environments, have been included in the revised manuscript. A mention on the consequent implications by the HVAC control viewpoint, i.e., that a larger humidification is required in winter and a smaller dehumidification is necessitated in summer, with a related increase of the heating costs in winter and a decrease of the cooling costs in summer, has also been added.
- The study performed by Casanova et al. has been mentioned in the text and included in the reference list of the revised manuscript.
- English language has been revised and improved where possible.
Reviewer 2 Report
Dear authors,
you did a really good review, or state of art and took profit of data in literature. I found this great. It is a good idea using enthalpy. However, as you also say, you canot generalise nor do any statistics because you really have a poor dataset.
I have some concerns:
1- You define VLS1h, but do not calculate it from data as is. I do not find this clear
2- When you write equation(2), AH calculation, you name a variable p, but never define it (yes, it is atmospheric pressure, but you should state it is). The problem with your data is that not always you can know the pressure at what experiments where done. By the way, what units do you use?
Otherwise, when using the standard equation https://carnotcycle.wordpress.com/2012/08/04/how-to-convert-relative-humidity-to-absolute-humidity/
you do not get results you say in table 1.
3- Table 1 does not show uniquely experimental data extracted from literature. You calculate some of its columns. By the way, I have not been able to find out VLS1h in Van Doremalen and Prussin references.
3-Equation (3) is equation (1) in numbers (fixed values).Then to obtain equation (4), what pressure did you use, what saturated vapor pressure? How did you do the division? How can you make RH "disappear" from the denominator?
4- in line 160 there is (6,). It should be (6)
5- reference (58) incomplete
Author Response
The authors wish to thank the Reviewer for her/his valuable comments and suggestions.
The manuscript has been modified accordingly, including some additional changes useful to improve the clarity of the paper.
The answers to each point raised by the Reviewer are reported below.
- The parameter VLS1h, defined as “percentage of detected levels of virus survival after one hour from spread”, has been calculated as the ratio between the viral load after one hour and that at the moment of its diffusion, the viral load being expressed indifferently in terms of PFU or TCID50. Paragraph “2.4 A parameter to correlate literature data” has been therefore rephrased, and more details from the cited literature have been included to allow an easier identification of the starting and ending points of the elaboration of data and its rationale.
- Variable p in Equation (2) is the total pressure of moist air, typically the atmospheric pressure in Pascal, as stated in the revised manuscript. Additionally, the empirical formula derived by Hyland and Wexler in 1983 (ASHRAE Trans. 89) to evaluate the saturated vapor pressure in Pascal as a function of temperature has also been included in the revised manuscript.
- The caption of Table 1 has been changed in “Virus survival obtained from literature experimental data and related information”. As far as the calculation of VLS1h is concerned, we included a brief explanation in the revised manuscript. In particular, the values of the relative infectious ratio reported by Prussin et al. for the Phi6 survival after 1 hour, expressed in terms of the ratio between the concentration of PFU derived from exposed samples and the concentration of PFU in the control sample, can be confronted with the other data from the literature that are expressed as the ratio between the virus TCID50 after 1 hour from spread and the virus titer TCID50 at the moment of its diffusion, since we are considering ratios and not absolute values and both these units (PFU and TCID50) refer to an infectious “dose” and, in the reference work they are used to evaluate infectious dose reduction over time, that is what is expressed with the VLS1h parameter. As regards the other three studies, in which data or information related to the time-distribution of TCID50 are reported as a function of RH and temperature, the parameter VLS1h has been calculated based on the estimation of the values of the virus titer TCID50 after 1 hour and at the moment of the virus dispersion.
- The “simplified” Equations (3)-(5) have been replaced by a more accurate equation, renumbered as Equation (4) in the revised manuscript, in which we explained how to use it to construct the diagram depicted in Figure 5.
- The full stop in reference (6) has been erased.
- The complete reference (58) has been incorporated in the revised manuscript.
Round 2
Reviewer 2 Report
Dear author,
thanks for changing equations (3) and (4), still
table (1) is badly calculated
when using the standard (valid nowadays, not in antique references) equation https://carnotcycle.wordpress.com/2012/08/04/how-to-convert-relative-humidity-to-absolute-humidity/
you do not get results you say in table 1.
In line 349 you say" which a well defined wet-bulb temperature twb is related to" Where do you define the relationship between enthalpy and wet bulb temperature? I can find a well defined relationship between wet and dry bulb tempèrature and humidity. Not enthalpy. Do you have any reference about it? From lines 304-306 you just say from literature. Nothing else.
Your equation (3) does not give appropriate results. Take any of the ones given
https://en.wikipedia.org/wiki/Vapour_pressure_of_water
On top of all these considerations, I do not understand why, if you change your equations, your results do not.
Author Response
The authors wish to thank the Reviewer for her/his valuable comments and suggestions.
The manuscript has been modified accordingly.
The answers to each point raised by the Reviewer are reported below.
- A reference for equations (1)-(2) has been included in the revised manuscript at line 261. Additionally, the units and values to be used in equation (2) have been indicated at lines 267-271, whereas the exact values of constants C1-C6 of equation (3) have been specified at lines 279-280. Notice that the absolute humidity AH, also called the humidity ratio, is normally defined as the ratio of the mass of water vapor to the mass of dry air in the moist air sample, i.e., the ratio mv/ma expressed in kgv/kgdry-air (sometimes in gv/kgdry-air), whereas the equation used in the website “carnotcycle” is referred to an absolute humidity defined as the mass of water vapor (in grams) per unit volume of moist air (in m3), which undoubtedly delivers a useful information, but it is not the quantity normally used to calculate the specific enthalpy of moist air.
- As regards the correlation between wet-bulb and dry-bulb temperatures rather than between wet-bulb temperature and specific enthalpy, the reviewer is perfectly right. Indeed, we used a typical approximation of HVAC designers, assuming that the wet-bulb temperature of an equilibrium state can be calculated as the temperature of the saturated state having the same specific enthalpy of the considered state, i.e., the thermodynamic state corresponding to the intersection between the isenthalpic and the saturation lines. However, since the knowledge of such an approximated “wet-bulb temperature” used in parallel with the specific enthalpy value does not add any further information, we preferred to avoid to mention it in the revised version of the manuscript, thus following the suggestion of the Reviewer.
- The Hyland-Wexler empirical relationship reported in equation (3) is generally considered more accurate than past equations, such as those listed in the website “wikipedia”. We checked the values, which well agree with those reported in saturated water tables. Attention should be addressed to the first member of the equation, which does not express the saturated vapor pressure in Pascal, but the natural logarithm of the saturated vapor pressure in Pascal.
- Both Table 1 and Figure 5 have been updated using the new and more accurate values.